# The *Eucalyptus grandis* chloroplast proteome: Seasonal variations in leaf development

**Amanda Cristina Baldassi, Tiago Santana Balbuena** *

Department of Agricultural, Livestock and Environmental Biotechnology, School of Agriculture and Veterinary Sciences, São Paulo State University (UNESP), Jaboticabal, São Paulo, Brazil

* tiago.balbuena@unesp.br

## Abstract

Chloroplast metabolism is very sensitive to environmental fluctuations and is intimately related to plant leaf development. Characterization of the chloroplast proteome dynamics can contribute to a better understanding on plant adaptation to different climate scenarios and leaf development processes. Herein, we carried out a discovery-driven analysis of the *Eucalyptus grandis* chloroplast proteome during leaf maturation and throughout different seasons of the year. The chloroplast proteome from young leaves differed the most from all assessed samples. Most upregulated proteins identified in mature and young leaves were those related to catabolic-redox signaling and biogenesis processes, respectively. Seasonal dynamics revealed unique proteome features in the fall and spring periods. The most abundant chloroplast protein in humid (wet) seasons (spring and summer) was a small subunit of RuBisCO, while in the dry periods (fall and winter) the proteins that showed the most pronounced accumulation were associated with photo-oxidative damage, Calvin cycle, shikimate pathway, and detoxification. Our investigation of the chloroplast proteome dynamics during leaf development revealed significant alterations in relation to the maturation event. Our findings also suggest that transition seasons induced the most pronounced chloroplast proteome changes over the year. This study contributes to a more comprehensive understanding on the subcellular mechanisms that lead to plant leaf adaptation and ultimately gives more insights into *Eucalyptus grandis* phenology.

## Introduction

Chloroplast is a subcellular compartment in which solar radiation is used to generate sugar compounds in a complex process named photosynthesis. It is also hosts several other metabolic processes that are essential for plant survival and is composed of inner and outer membrane, stroma, thylakoidal membrane, and lumen [1]. Moreover, *Eucalyptus* leaves have a great amount of antioxidants, particularly flavonoids, which provide protection against from ROS damage, oxidative stress and can increase stress tolerance [2]. Plastid genome encodes 120–130 proteins, however approximately 2000 proteins, are synthesized in the cytosol, and imported into the organelle through the envelope membranes [3, 4]. In *Eucalyptus grandis*, the chloroplast genome presents approximately 160 kb [5, 6].

**Data Availability Statement:** All mass spectrometric data are available via ProteomeXchange under identifier PXD029004.

**Funding:** This work had financial support from São Paulo Research Foundation - FAPESP (grant

number 2018/15035-8). ACB also received a scholarship from the same foundation (scholarship number 2019/12580-8). TSB received a scholarship from CNPq (304479/2020-9). Research funders had no role in study design, data collection and analysis, decision to publish, or preparation of the manuscript and authors.

**Competing interests:** The authors have declared that no competing interests exist.

As chloroplasts are very sensitive to environmental fluctuations, dynamic proteome changes may represent a key factor for understanding their responses to abiotic stresses [7]. The most recent chloroplast proteomics studies have aimed to identify (1) changes induced by biotic and abiotic stresses [8–12], (2) mechanisms of remodeling and reprogramming of plant metabolism [13, 14], (3) biogenesis and senescence [15–17], (4) chloroplast development [18, 19], in addition to (5) novel techniques for increasing proteome coverage [20–22].

Chloroplast biogenesis starts from proplastids, which, upon exposure to light, turn into mature chloroplasts with functional electron transport carriers, membrane channel proteins, and stromal enzymes that are able to capture, assimilate, and storage carbon [23]. During senescence, they differentiate from gerontoplasts, which lead to leaf yellowing, necrosis, and death [24]. Thereby, plastid development is intimately related to the development of the whole plant at different stages [25]. The manipulation of several agronomic traits, *e.g.*, photosynthetic rate, stomatal conductance, chlorophyll content, and PSII photochemical, has been reported to delay the senescence process and increase plant biomass, seed yield, seed protein composition, and tolerance to abiotic stresses [26–28].

Plants are sessile organisms, which means that they are subjected to seasonal climate alterations and adverse environmental conditions. Seasonal climate variations can drive plant growth and development, favoring a metabolic adaptation of the life cycle with changes in the environment [29]. Strong seasonal effects have been observed on the growth of perennial plants due to varying availability of water, light, nutrients, and temperature [30, 31]. In *Populus* species, for instance, seasonal phenology is largely related to changes in growth and development, such as bud flush in spring followed by vegetative growth in the summer; while during the fall, the trees present growth inhibition and bud formation, followed by leaf senescence, abscission, and cold acclimation before the winter season [32]. Regarding photosynthetic physiological changes, studies have revealed that the quantum efficiency seasonal pattern shows minimal values in the spring, increasing during the summer, and reaching maximal values in the fall followed by a decrease in the winter [33, 34].

Understanding functional and regulatory networks based on leaf development process and seasonal metabolic changes can provide valuable information for increasing plant productivity and learning the mechanisms that lead to plant adaptation. This is the first exploratory investigation on proteome changes during leaf development and seasonal variations in *Eucalyptus grandis* young plants on an organellar level. This research aimed to carry out an exploratory analysis of chloroplast alterations during leaf development and seasonal variations in the *Eucalyptus grandis* chloroplast proteome.

## Materials and methods

### Plant cultivation and sampling

*Eucalyptus grandis* plants were planted at the "Horto Florestal" of the São Paulo State University (Jaboticabal–SP, Brazil) using a randomized block design experiment with 2m x 3m spacing. Fertilization and irrigation were carried out twice a month and three times a week, respectively. The region climate is defined as tropical and characterized by rainfalls concentrated in the summer and dry winter [35]. Agrometeorological data were collected daily at the UNESP environmental station (S1 Table).

For the leaf developmental assay, three biological replicates, comprising five randomly selected branches, were divided into three different regions (young, middle, and mature), according to the fluorescence data (FV/FM) and chlorophyll relative quantification (CCI), and used to assess proteome changes throughout development (S1A Fig). For the chloroplast proteome seasonal variation assay, four biological replicates, comprising forty leaves from the first

until the fifth node, were assessed in all four seasons from August/2019 to August/2020 (S1B Fig).

## Fluorescence measurement and chlorophyll relative quantification

Fluorescence analysis was performed using the fluorometer FluorPen FP 100 (Photon Systems Instruments) assessing the adaxial surface of leaves from different regions of the 15 branches. The fluorescence values obtained directly from the device were used to assess the potential quantum efficiency of photosystem II (FV/FM). Relative chlorophyll was quantified on CCM-200 equipment (Opti-Sciences), and the adaxial surface of leaves from different regions of the 15 branches was examined. The CCI values obtained directly from the device were used to assess relative chlorophyll quantification. Statistical analysis was performed according to the ANOVA test ($p < 0.05$), and Tukey's test was used for comparison among different leaves.

## Chloroplast proteome extraction

Six grams of leaves were collected, fine sliced, and transferred to tubes containing 20 mL of chloroplast isolation buffer (CIB, CP-ISO-Sigma Aldrich). The leaf tissue was homogenized (Omni GLH Turrax) and filtered through a nylon membrane. The processed material was centrifuged, and the chloroplasts were isolated in an 80%/40% (1:2 v/v) Percoll density gradient. After washing, the chloroplast pellet was resuspended in a protein extraction buffer (Thiourea 2M; CHAPS; Tris-HCl 1M, pH 8; Glycerol 50%; Triton X-100; DTT 1M and Urea 8M) and the lysis of intact chloroplasts was carried out through 20 cycles of sonication. After centrifugation, the supernatant was precipitated in cold acetone for 16 hours at -30˚C, and the protein pellet obtained was solubilized in Tris buffer (125 mM Tris pH 6.8, 20% glycerol, 1% SDS, 0.1% DTT).

## Total leaf proteome extraction

To assess the enrichment of the chloroplast subproteome in relation to the total leaf proteome, intact leaves were frozen in -80˚C and grinded in Tris extraction buffer (125 mM Tris, pH 6.8, 20% glycerol, 1% SDS, 0.1% DTT). Upon centrifugation at 4˚C for 15 min, the supernatant was transferred to a clean tube and the proteins were precipitated in cold acetone for 16 hours at -30˚C. Finally, the proteins were solubilized in Tris buffer before protein quantification assay.

## Protein quantification assay and SDS-PAGE

Sample quantifications followed the Bradford method using a BSA curve as reference. Aliquots of 50 µg of proteins were separated for 1 hour by denaturing electrophoresis on a polyacrylamide gel containing sodium dodecyl sulfate (SDS-PAGE) to obtain a single band. The electrophoretic system was composed of a stacking gel (5%) and a separation gel (12.5%). The single protein band was excised from the gel and subjected to in-gel digestion.

## Mass spectrometry analysis

Proteins were in-gel digested using trypsin at 1:10 proportion (trypsin:protein) [36]. To clean up the peptide samples, extracts were filtered in an acetate cellulose membrane (Spin-X, Corning) by centrifugation for 7 minutes. The flow-through was collected using a clean tube, concentrated, and then desalted with ZipTip Pipette Tips (Millipore) according to the manufacturer's protocol. The amount of 1ug of extracted peptides was analyzed through liquid chromatography coupled to a tandem mass spectrometer. Tryptic peptides were resuspended

in $H_2O$ containing 0.1% formic acid and separated in an EASY-nLC1000 chromatographic system (Thermo Fisher Scientific) for 95 minutes using a C18 nano-column under a constant flow of 400 nL/min and increasing concentrations of acetonitrile (7 to 30% over 70 min and then 70% in 25 min). The peptides were ionized through electrospray (3.5 kV) and the mass spectrometer (Q-Exactive, Thermo Fisher Scientific) was operated in data-dependent acquisition mode according to a top 20 acquisition method with dynamic exclusion time adjusted to five seconds. Full scans and MS/MS scans were acquired with resolutions equal to 70,000 and 35,000 FWHM (full width at half maximum), respectively. The MS/MS spectra were generated from HCD fragmentation (high-energy collision dissociation) of the precursor peptide ions isolated under 35 eV collision energy. Mass spectrometric data are available via ProteomeXchange under identifier PXD029004.

## Protein identification

The peptides and protein inferences were identified through stringent searches of experimental data against the protein database *Eucalyptus grandis* v.2.0 (http://www.phytozome.net/) on MaxQuant (v. 1.6.3.3) [37] based on dedicated search parameters: 20 ppm tolerance for precursor ions, oxidation of methionines and carbamidomethylation of cysteines–selected as dynamic and static modifications, respectively. Maximum missed cleavages of 2 and false-discovery rate adjusted to 1%. Relative abundance of the chloroplast proteins were quantified based on the label free-based approach (LFQ intensity). Relative abundance of total leaf proteins was obtained following the NSAF ("normalized spectral abundance factor") approach [38]. Mass spectrometry proteomics data are currently stored on the ProteomeXchange Consortium via PRIDE partner repository under dataset identifier PXD029004.

## Statistical analysis and functional annotation

Data processing and statistical analysis were carried out on the Perseus software (v. 1.6.15.0) [39]. Firstly, all contaminants and reverse sequences were removed from data analysis. Subsequently, label-free quantification data (LFQ) were transformed into $log_2$ scale, and a protein was considered a true hit if identified in at least two out of three replicates. Missing values imputation was carried out from the normal distribution curve from each biological replicate. The means of ANOVA test and Tukey comparison test ($p < 0.05$) were used in the comparative analyses. Multivariate analyses were carried out with differentially abundant proteins for both assays on the Perseus software. For PCA and hierarchical clustering, data were z-score normalized. Hierarchical clustering was performed using Euclidean distance and complete linkage.

Functional annotation of the differentially abundant proteins was carried out using the Blast2GO platform ($p < 0,05$) [40]. The ten most representative level-four biological processes were considered in the functional characterization aiming to balance between a comprehensive overview and detailed information on the targeted biological process. Level four was chosen for including photosynthesis, an important process in the context of leaves and chloroplasts.

## Subcellular prediction

*In silico* analysis was performed using predictors of subcellular location and chloroplast proteome repositories to assess and confirm the cellular location of the identified proteins. For this purpose, a protein was considered plastidial if identified by at least one of the subcellular prediction algorithms–ChloroP [41] and Predotar [42]–or by presenting at least one homologous protein in any of the databases used–PPDB [43] or AT-Chloro [44].

## Results and discussion

Before performing any description or comparative proteome analysis, we decided to verify the efficiency of the chloroplast enrichment strategy used herein. For this purpose, we compared the abundance levels, in terms of NSAF and LFQ intensities, of a well-known chloroplast proteoform Eucgr.C03525.1 (Ribulose-bisphosphate carboxylase–RBCL) against five potential mitochondrial contaminants commonly found in chloroplast extracts (Table 1). Quantitative analysis indicated that the chloroplast proteoform was enriched upon the extraction process, whereas the mitochondrial proteoforms presented a decrease in the abundance levels. These results indicated that our chloroplast extraction and enrichment strategy was effective in reducing the cross-contaminants protein abundance, as well as in the enrichment process of chloroplast-related proteins in the target samples.

### Photosynthetic proteins comprise most of the *Eucalyptus grandis* chloroplast proteome

Aiming to illustrate a general overview of the chloroplast proteome isolated from *Eucalyptus grandis* leaves, we compiled the identification datasets from both leaf development and seasonal experiments. The combined analysis resulted in a stringent, non-redundant dataset of 431 chloroplast proteins (S2 Table). Gene ontology analysis indicated that most of the isolated proteins were involved in the "organonitrogen compound metabolic process", "organic substance biosynthetic process", and "macromolecule metabolic process" (Fig 1A). Although these biological processes were the most prominent in terms of protein identifications, an MS1-based quantitative analysis indicated that the "photosynthesis process" represented 11% of the isolated chloroplast proteomes (Fig 1B). The fraction of "photosynthetic process" related to proteins becomes even more pronounced within the isolated proteome upon the analysis of the most abundant proteins identified in the entire experiment (Fig 1C). The RuBisCO small chain (Eucgr.J01502.2.p) was the most abundant protein identified herein, showing approximately 4000 MS/MS counts. The proteins involved in the photosystem II structure were also highly abundant: two extrinsic proteins (PsbO-2 –Eucgr.I01025.1.p and PsbQ-2 –Eucgr.D00854.1.p) that form the oxygen-evolving complex (OEC) and a light-harvesting complex

**Table 1. Enrichment analysis of chloroplast proteins in *Eucalyptus grandis* using a comparative approach between total leaf protein extract and chloroplast protein extract.**

| Description[a] | Loc[b] | NSAF[c] | | | | LFQ intensity[d] | | | |
|---|---|---|---|---|---|---|---|---|---|
| | | Leaf | Chloro | FD | | Leaf | Chloro | FD | |
| **Contaminants** | | | | | | | | | |
| Malate dehydrogenase (Eucgr.F01209.1) | M | 0.0036 | 0.0010 | 3.52 | ↓ | 3.E+08 | 2.E+07 | 15.39 | ↓ |
| Isocitrate dehydrogenase (Eucgr.A01135.1) | M | 0.0010 | 0.0005 | 2.27 | ↓ | 2.E+08 | 1.E+07 | 10.98 | ↓ |
| Citrate synthase (Eucgr.G03412.1) | M | 0.0004 | 0.0000 | 10.98 | ↓ | 3.E+07 | 2.E+06 | 16.04 | ↓ |
| Fumarase (Eucgr.H01081.1) | M | 0.0001 | 0.0000 | 3.65 | ↓ | 7.E+06 | 4.E+05 | 20.04 | ↓ |
| Succinate dehydrogenase (Eucgr.F00243.1) | M | 0.0002 | 0.0002 | 1.02 | ↓ | 2.E+06 | 2.E+06 | 1.13 | ↓ |
| **Positive control** | | | | | | | | | |
| Ribulose-bisphosphate carboxylase-large (Eucgr.C03525.1) | C | 0.0155 | 0.0276 | 1.78 | ↑ | 3.E+09 | 3.E+09 | 1.19 | ↑ |

Local: Localization, M: Mitochondria, C: Chloroplast, Leaf: Total leaf protein extract, Chloro: Chloroplast protein extract, FD: Fold-Change.

[a] Functional Annotation and protein identifier according to the *E. grandis* v 2.0 annotation

[b] Subcellular protein location

[c] Relative protein expression by the NSAF approach

[d] Relative protein expression by the LFQ intensity approach

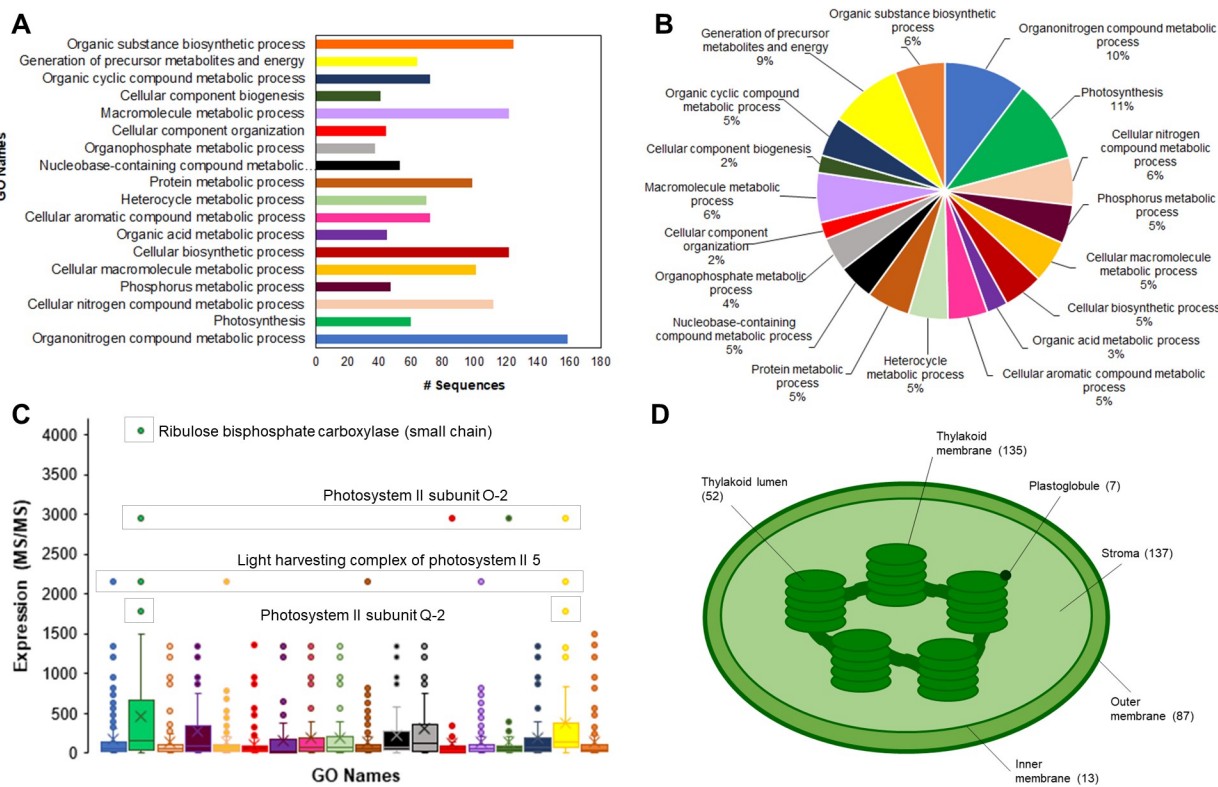

**Fig 1. The *Eucalyptus grandis* chloroplast proteome.** Classification analysis, in terms of the number of identified proteins (A) and terms of the percentage of identified proteins (B), upon categorization using the "biological process" ontology; protein expression analysis of the Gene Ontology terms (C) and number of protein identifications per chloroplast sub-cellular compartment (D).

protein from the PSII (Eucgr.F04099.1.p) corresponding to 1500 to 3000 MS/MS counts (Fig 1C).

Regarding the suborganellar localization of the 431 chloroplast proteins, 137 proteins were present in the stroma compartment, whilst 135 were thylakoidal proteins. A total of 87 proteins were predicted *in silico* to be located in the chloroplast outer membrane, and 13 identifications were found to be part of the chloroplast inner membrane. Finally, 52 proteins were located in the chloroplast lumen and 7 proteins herein identified had been previously reported as part of the plastoglobule structure (Fig 1D).

## Chloroplast proteome reflects the physiological status in developing leaves

Based on the varying FV/FM ratios and the differences observed from the relative chlorophyll quantitation analysis (S2 Fig), we isolated the chloroplast proteome from young, middle, and mature leaves. Quantitative data from the isolated proteins were used to demonstrate the changes within this subcellular structure throughout the leaf development process. Pairwise analysis indicated that 61 proteins were considered to be differentially abundant over the development process, with the young chloroplast proteome as the most dissimilar of the three different regions (Fig 2A). Such result is in accordance with physiological evidence highlighted in S1 Fig, as the young leaf tissue presented the most significant variations concerning middle and mature regions. A study on sugarcane leaves have also demonstrated that physiological parameters, such as chlorophyll content, nitrogen content, and RubisCO activity, were greater and more consistent between mature and transition tissues than in young regions [45].

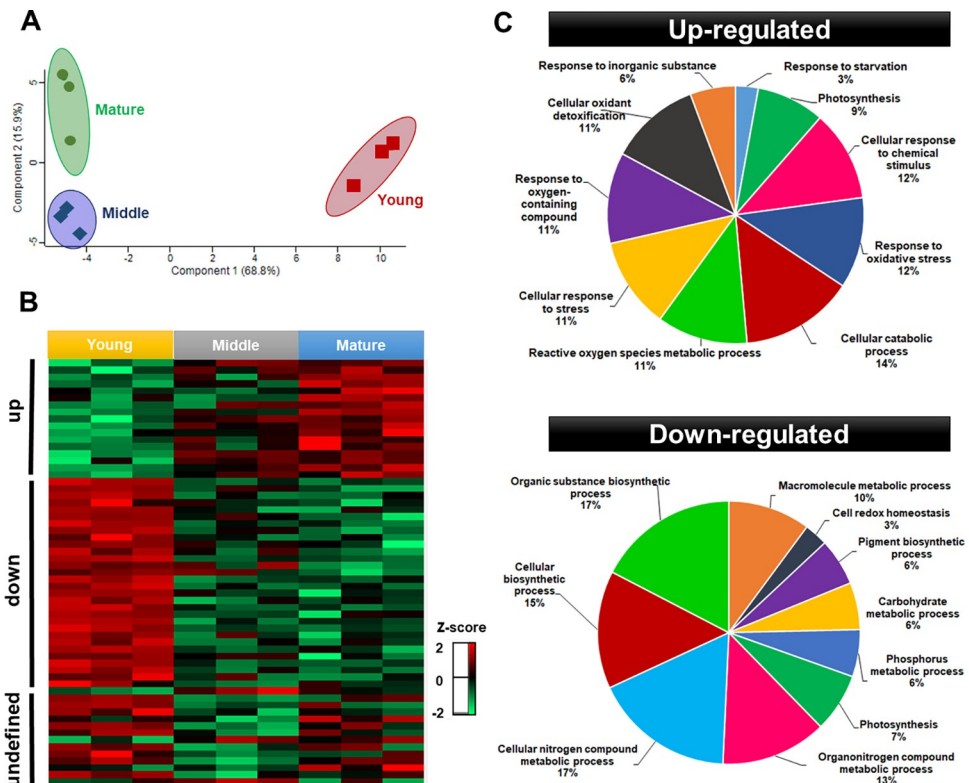

**Fig 2. Multivariate analyses of the 61 differentially abundant chloroplast proteins isolated from young, middle, and mature *Eucalyptus grandis* leaves.** Principal component analysis (A), clustering expression analysis (B) and functional categorization of the differentially-regulated chloroplast proteins between young (downregulated) and mature (upregulated) (C). The z-score indicates the distance of the data compared to the mean value.

Clustering analysis allowed to classify the differentially abundant proteins (DAP) in three distinctive groups: upregulated– 17 proteins with higher abundance over leaf development, downregulated–a set of 29 proteins with lower abundance along the studied process, and an undefined group, comprising 15 proteins with an unclear abundance pattern (Fig 2B).

Proteins with higher abundance throughput leaf development comprised those involved in the antioxidant metabolism (APX4, CAT2, tAPX, and SODFe2), stress responses/protection against lipid peroxidation (IDH-V, MPC1, and CHL), and photosynthesis (PSBQ-2, PSBO-2, GAPB, and FTRA2). Additionally, proteins involved in post-transcriptional modifications (CBR), regulation of PSII (TLP), amino acid metabolism (PYD2), photorespiration (GCT), carboxylation, and $CO_2$ diffusion (CA1) and plastoglobuli development (FIB4) were also found to be upregulated. Conversely, ribosomal proteins (RPL13, PRPL1, RBD1, RBD2, and RPS1), proteins related to different biosynthetic processes (CRD1, PORA, THI1, PLP and IPP2, GSA1, LPD1, and FKB), folding, modification and transport (TPR, HSC70-2 and CLPC1, CURT1C, ENR1, LTP1, and LTP2), antioxidants and detoxification (TPX, AOR 1 and AOR2) and also photosynthesis proteins (RCA, RBCS1, RBCS2, and PSBS) were observed to lower abundance during leaf development. Proteins with unclear abundance pattern or those presenting a transient increase or decrease in the abundance levels were found to be involved in the photosynthetic process and electron transport chain (LHB1, PSAF, and THOL), in addition to folding, modification, and transport (HSP70, HSP21, ENR2, OEP24A, and RABE1B) and biosynthetic processes (NDPK1, MEE32, and PSAT). Proteins responsible for peptide

degradation/repair of photosynthetic machinery (PREP2 and FtsH), oxidoreductase, and protease (CHLP and ASP) were also clustered within such heterogeneous group (S3 Table).

Aiming to highlight changes in the chloroplast metabolic processes along leaf development, up and downregulated proteins were cross-referenced to corresponding gene ontology terms. Functional categories revealed differences between the chloroplast protein groups (Fig 2C). Leaf development induced "cellular catabolic processes" in *Eucalyptus grandis* chloroplasts for being the most representative category within the upregulated group. Responses to stress, mainly to oxidative stress and cellular detoxification, were the most abundant functional category in upregulated throughout leaf development, based on the maturity of the photosynthetic apparatus and the higher photosynthesis rates in young leaves. In non-stressing conditions, mature leaves are expected to maintain cellular homeostasis with a dynamic balance between the light energy absorbed and energy consumption [46]. In mature leaves, generation of reactive oxygen species and changes in the redox state in the photosynthetic electron transport chain are important indicators of operational retrograde signals. The function of operational retrograde signaling differs from biogenic signaling as it focuses on the adjustments and cellular homeostasis of the chloroplast in response to different perturbations and developmental cues [47].

Conversely, the most representative functional class identified in the chloroplast downregulated protein group (*i.e.*, abundant in young leaves) were involved with cellular and organic biosynthetic processes, being the most representative in the "organic substance biosynthetic process" and the "nitrogen compound metabolic process". These results are under the developmental stage of the young leaves used here. As plastids are intimately related to developmental and environmental signals, chloroplasts from young leaves were expected to reflect the biogenesis processes, which involves large-scale gene transcription followed by synthesis, importing, and assemblies of biogenesis proteins, such as those involved in plastid translation apparatus and enzymes in the synthesis of isoprenoids and tetrapyrroles [48]. In turn, in mature leaves, redox-related proteins comprised most of the differentially regulated proteins; the only category suggested by gene ontology analysis in young leaves for "cell redox homeostasis" comprised only 3% of all downregulated proteins. A study on the developmental dynamics of maize leaves also found upregulated genes that encode enzymes involved in diverse biosynthetic processes in young leaves. Similarly, in mature leaves, the upregulated genes predominant encoded Calvin cycle, redox regulation, and the light reactions of photosynthesis [48].

For better insights into the changing patterns of the up and downregulated proteins, we classified the chloroplast proteins into three expression trends, as shown in Fig 3. The early responsive proteins comprised the subset with a drastic change in the abundance level in middle leaf tissues, remaining at similar levels in both mature and young leaves. Conversely, late responsive proteins showed a drastic abundance change only in the contrasting leaf tissue (young or mature leaves). We considered as almost linear responsive the proteins without drastic changes throughout the leaf development process, but with a significant change between mature and young tissues.

Early upregulated proteins (*i.e.*, early increased abundance along leaf development) were identified as follows: Mitochondrial pyruvate carrier (MPC1), Isocitrate dehydrogenase V (IDH-V), Ferredoxin/thioredoxin reductase subunit A2 (FTRA2), Pyrimidine 2 (PYD2), Carbonic anhydrase 1 (CA1), and Glycine cleavage T-protein family (GCT). The most pronounced increase occurred in the Pyrimidine 2 (PYD2) protein from young to middle tissues. This protein acts in beta-alanine biosynthesis and nitrogen recycling from pyrimidines to general nitrogen metabolism [49]. Late upregulated proteins (*i.e.*, late increased abundance along leaf development) were identified as follows: Photosystem II subunit Q-2, Glyceraldehyde-

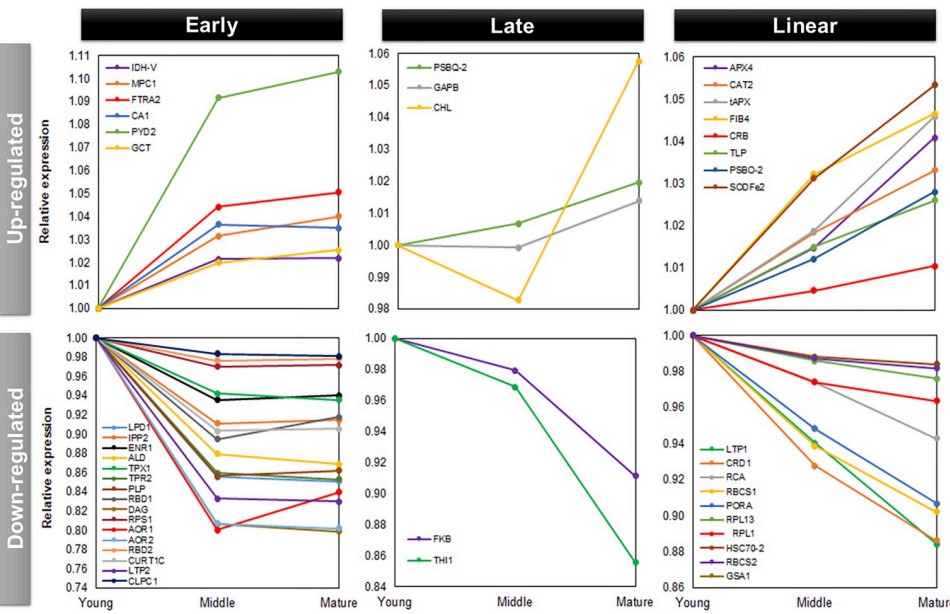

**Fig 3. Expression trends of the differentially regulated chloroplast proteins.** Upregulated proteins increased abundance throughout leaf development, while downregulated proteins decreased abundance during leaf development.

3-phosphate dehydrogenase B subunit, and Chloroplast lipocalin (PSBQ-2, GAPB, CHL). The chloroplast lipocalin (CHL) presented the most pronounced increase in abundance. This protein acts in the protection of thylakoidal membrane lipids against reactive oxygen species, especially singlet oxygen, induced by high light and drought stresses occurring mainly at early stress stages in *Arabidopsis* [50]. It has also been reported that stress conditions cause the CHL to protect the thylakoid membrane by facilitating a sustained non-photochemical quenching (NPQ) in the LHCII in *Arabidopsis* [51]. A nearly increasing linear responsive pattern throughout leaf development was found in the following chloroplast proteins: Ascorbate peroxidase 4 (APX4), Catalase 2 (CAT2), Thylakoidal ascorbate peroxidase (tAPX), Fe superoxide dismutase 2 (SODFe2), Thylakoid lumen 18.3 kDa protein (TLP), Photosystem II subunit O-2 (PSBO-2), Chloroplast RNA binding (CBR) and Plastid-lipid associated protein PAP/fibrillin (FIB4).

Early downregulated proteins (*i.e.* early decreased abundance along leaf development) was the largest trend group according to our categorization (16 proteins) comprising proteins involved in the biosynthesis of fatty acids, isoprenoids, and amino acids (LPD1, IPP2, and PLP), chloroplast RNA binding proteins (RBD1, RBD2, and DAG), ribosomal proteins acting in the biosynthesis of thylakoid membrane and thylakoid curvature (RPS1 and CURT1C), transport, modification and import into the chloroplast (LTP2, TPR, and CLPC1), proteins associated with detoxification (TPX1, AOR1, and AOR2), binding cofactors (ENR1), and sucrose metabolism (ALD). A nearly linear downregulated responsive trend appeared in 10 protein sequences showed that act in carbon fixation RubisCO (RCA, RBCS1, and RBCS2), chlorophyll biosynthesis (CRD1, PORA, GSA1), transport and folding (LTP1 and HSC70-2), and structural ribosomal proteins (RPL13 and RPL1). In turn, only two proteins showed a late downregulated pattern, pfkB-like carbohydrate kinase family protein and Thiazole biosynthetic enzyme, chloroplast (FKB and THI1).

In conclusion, while investigating the chloroplast proteome changes over leaf development, we found significant alterations in the target proteome regarding the maturation process. Out

of the 431 proteins isolated from the chloroplast material, 61 showed significant changes in the abundance throughout leaf development. There was no clear bias towards a unique change in the proteome composition; instead, we found distinct patterns of protein accumulation. In addition to pinpointing proteins with pronounced changes ranging the studied phenomenon, through a quantitative proteomics approach, we also specified the most particular biological and metabolic processes involving leaf development. However, as the proteome of any living entity is constantly changing, we also decided to investigate the undisturbed (*i.e.*, without inducing anthropogenic alterations due to external stimulus) sub-cellular proteome dynamics by analyzing the seasonal changes occurring in the chloroplast proteome of *Eucalyptus grandis* plants cultivated at the State of Sao Paulo Forest Nursery between September 2019 and August 2020.

## Transition seasons induce the most pronounced chloroplast proteome changes

Meteorological data revealed only small changes in the mean and maximum temperatures recorded throughout the study period, with the minimum diurnal temperature ranges observed during the summer season. Precipitation analysis revealed extremely dry fall and winter seasons and mean monthly rainfall of almost 10 mm over the summer (Fig 4A). Multi-variate analysis of the *E. grandis* chloroplast proteomes suggested that both the fall and the spring-related samples presented the most dissimilar profiles of the studied seasons (Fig 4B). This is likely to be related to the phenological transitions commonly occurring in the species during such periods. In *Populus* species, phenology is largely related to seasonal changes in growth and development, such as bud flush in spring followed by vegetative growth in summer. In turn, during the fall, the trees presented growth cessation and bud formation, followed by leaf senescence and abscission, in addition to cold acclimation for the winter [32]. Our comparative proteome analysis revealed that 41 proteins were differentially regulated across the seasons of the year in *E. grandis* (S4 Table).

Hierarchical clustering analysis of DAP proteins suggests 5 different clusters according to their abundance patterns (Fig 5A). As the transition seasons (fall and spring) presented the most dissimilar profiles, it is worth highlighting the clusters that showed the most expressive differences in these specific periods.

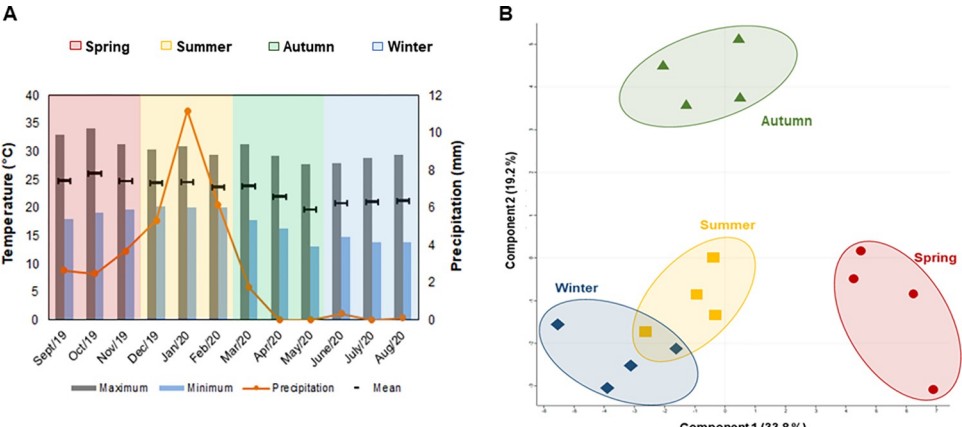

**Fig 4. Meteorological data recorded at the Sao Paulo State University Forest Nursery during *Eucalyptus grandis* cultivation.** Changes in temperature and rainfall rates between August 2019 and September 2020 (A) and multivariate analysis of the chloroplast proteome samples collected in the summer, fall, winter, and spring (B).

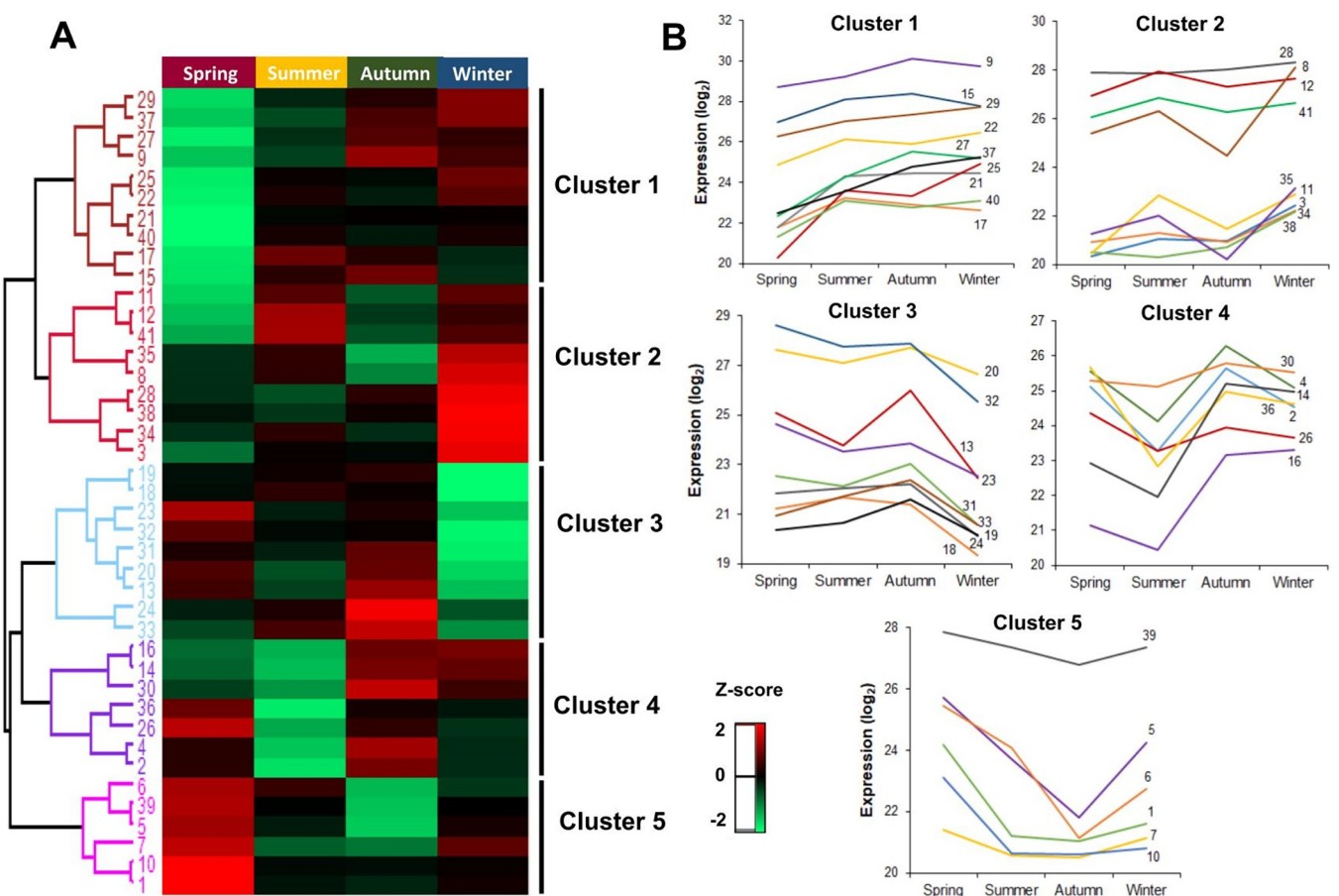

**Fig 5. Differentially abundant proteins identified in *Eucalyptus grandis* chloroplasts in different seasons of the year.** Hierarchical clustering (A) and protein abundance trends (B). The z-score indicates the distance of the data compared to the mean value.

Cluster 1 comprises 10 identifications with the lowest abundance values in the spring season. Proteins are related to light-harvesting complex, carbonyl detoxification, folding and modifications, carbohydrate biosynthesis, hydrolases, amino acid precursors, and ATP carriers. Conversely, Cluster 5 is composed of 6 proteins that showed higher abundance in the spring. Most of these identifications are associated with RuBisCO regulation to enhance photosynthetic capacity. Other identifications are responsible for lipid transport, protein modification, and translation (Fig 5B).

Other studies have revealed that quantum efficiency seasonal pattern has minimal values in the spring, increasing over the summer, reaching maximal values in the fall, and finally decreasing in the winter [33, 34]. Such scenario is demonstrated in Cluster 1, with proteins LHB1 and LHCB4, which participate in the first steps of photosynthesis and are very efficient in capturing light energy and transforming light into chemical energy [52]. Photosynthetic capacity in plants has also been reported to fluctuate with increased levels throughout leaf development, reaching its peak in the spring or early summer, and remaining stable or gradually declining over the summer [53–55]. Such pattern appears in Cluster 5, with RuBisCO-related proteins (RCA, RBCS1, and RBCS2). Leaf fluorescence and photosynthesis are expected to increase over the spring, especially in chlorophyll content. In the fall, these parameters are reduced when associated with leaf physiological properties such as leaf aging and senescence [56].

Cluster 4 presented the highest abundance for the fall season (Fig 5B), with 7 identifications associated with the following processes: PSII regulation, chlorophyll biosynthesis, translation, phospholipid transport; in addition to two histones that play an important role in regulating transcription and DNA repair and replication.

As previously described, trees undergo growth cessation and establish dormancy before the onset of winter over a transition season. Low temperatures were found to facilitate photoinhibition but not necessarily to cause it [57]; in addition, downregulation of Calvin-Benson cycle activity and gas exchange precedes the reorganization of photosynthetic apparatus in thylakoid membrane during the fall [58]. All these events are partially related to PSII regulation and corroborate our analysis, where we found an upregulated protein related to the regulation of PSII in Cluster 4. Furthermore, a plant epigenome is very responsive to changes in the environment, and variations in epigenetic silencing mechanisms are among the key factors that assist plants to adapt to different climate conditions [59]. DNA methylation and histone modifications associated with chromatin remodeling were reported to be the major changes occurring throughout dormancy season transitions [60]. The up-regulation of two histones in the fall occurring in Cluster 4 could be associated with these epigenetics events.

Aiming to investigate which chloroplast proteins had the most pronounced synthesis in each season, we carried out a fold-change analysis, as shown in Fig 6. The red boxes indicate proteins with significant change equal or higher than 2-folds, blue boxes show significant change equal or higher than 5-folds, and white boxes point to proteins with change lower than 2-folds.

The most significant abundance increase had in both wet seasons (spring and summer) occurred in the RBCS1 protein. This is one of the small subunits of the RuBisCO enzyme, whose down-regulation has been observed in *Cistus albidus* [61] in drought periods, such as fall and winter. In *Lolium perenne*, the RbcS transcripts reached the highest levels over the spring, gradually decreasing between the summer and winter [62]. Similarly, all photosynthetic parameters measured in *Quercus coccinea Muench* (scarlet oak) showed a peak in the late summer, followed by a sharp decline in the fall [63].

In contrast, the most abundant proteins in both dry seasons (fall and winter) were MPH1, PORA, MEE32, TPI and AOR. MPH1 (proline-rich family protein) is a PSII-associated proline-rich protein that participates in the maintenance of normal PSII activity and protection of PSII from photooxidative damage under excessive light conditions [64]. The PORA protein (Protochlorophyllide oxidoreductase A) catalyzes the reduction of protochlorophyllide to chlorophyllide, subsequently producing chlorophyll a and b [65, 66]. In addition to its role in chlorophyll biosynthesis, PORA can also exert a photoprotective action in the transition from dark to light [67], whereas the *Triose phosphate isomerase* enzyme (TPI) interconverts glyceraldehyde 3-phosphate (G3P) in the Calvin cycle placed in chloroplasts. Up-regulation of TPI in the winter has been reported in experiments performed on tubers and *Eucalyptus grandis* bark [68, 69].

The bifunctional enzyme MEE32 (dehydroquinate dehydratase, putative/shikimate dehydrogenase) is responsible for steps three and four of the shikimate pathways, catalyzing the dehydration of dehydroquinate to dehydroshikimate and the reversible reduction of dehydroshikimate to shikimate, respectively. Transgenic lines of tobacco with suppressed MEE32 gene displayed severe growth retardation and reduced content of aromatic amino acids and downstream products such as lignin and chlorogenic acid [70]. In *Eucalyptus camaldulensis*, it confers aluminum tolerance [71]. AOR protein (Oxidoreductase, zinc-binding dehydrogenase family protein) detoxify α, β-unsaturated carbonyls by reducing a highly electrophilic α, β-unsaturated bond using NAD(P)H [72, 73]. It favors the detoxifying of stromal lipid peroxide-derived reactive carbonyls (RCs) produced under oxidative stress in chloroplasts [73], in addition to protecting dark respiration and supporting plant growth during the night [74].

| ID | Accession | Description | Humid seasons | | Dry seasons | |
| --- | --- | --- | --- | --- | --- | --- |
| | | | Spring | Summer | Autumn | Winter |
| 1 | Eucgr.A00746.1 | LTP1 | >5 | | | |
| 2 | Eucgr.K03160.1 | HTA12 | | | >5 | 2 |
| 3 | Eucgr.A02448.1 | ENR1 | | | | 2 |
| 4 | Eucgr.H04433.2 | G-H2AX | | | 2 | |
| 5 | Eucgr.B02310.1 | RCA | >5 | 2 | | |
| 6 | Eucgr.B03013.1 | RBCS1 | >5 | >5 | | |
| 7 | Eucgr.C01875.1 | ENR2 | | | | |
| 8 | Eucgr.C02720.2 | ASP5 | | 2 | | >5 |
| 9 | Eucgr.D00322.1 | LHB1 | | | 2 | 2 |
| 10 | Eucgr.D00609.2 | RPS5 | >5 | | | |
| 11 | Eucgr.D02023.1 | SCO1 | | | | >5 |
| 12 | Eucgr.D02467.1 | TKL | | | | |
| 13 | Eucgr.E02606.1 | PETA | >5 | 2 | 2 | |
| 14 | Eucgr.F01098.1 | PORA | | | >5 | >5 |
| 15 | Eucgr.G03060.1 | LHCB4.3 | | | 2 | |
| 16 | Eucgr.G03205.2 | MPH1 | | | >5 | >5 |
| 17 | Eucgr.G03388.1 | AAC2 | | | 2 | |
| 18 | Eucgr.H00243.3 | CLPD | 2 | >5 | | |
| 19 | Eucgr.H01473.3 | RRF | 2 | 2 | | |
| 20 | Eucgr.H02991.1 | APTF | 2 | | | |
| 21 | Eucgr.H04427.1 | MEE32 | | | >5 | >5 |
| 22 | Eucgr.H04673.1 | GAPC2 | | | 2 | 2 |
| 23 | Eucgr.I00441.1 | PRPL29 | 2 | | | |
| 24 | Eucgr.I00922.1 | ZEP | | | 2 | |
| 25 | Eucgr.I01374.1 | TPI | | | >5 | >5 |
| 26 | Eucgr.I01643.1 | RPL27 | | | | |
| 27 | Eucgr.I01808.1 | AOR | | | >5 | >5 |
| 28 | Eucgr.I02771.1 | HSC70-2 | | | | |
| 29 | Eucgr.J00025.1 | HSC70 | | | 2 | 2 |
| 30 | Eucgr.J00865.1 | HCF136 | | | | |
| 31 | Eucgr.J02666.1 | RPL14 | 2 | 2 | | |
| 32 | Eucgr.J02738.1 | ATPB | >5 | 2 | | |
| 33 | Eucgr.K00402.3 | TPR | | 2 | 2 | |
| 34 | Eucgr.K00732.1 | PDIL2-2 | | | | 2 |
| 35 | Eucgr.K01105.1 | AHRI | 2 | 2 | | 2 |
| 36 | Eucgr.K01283.1 | LTP2 | 2 | | 2 | 2 |
| 37 | Eucgr.K01490.1 | ENR3 | | | 2 | >5 |
| 38 | Eucgr.K01713.1 | SFGH | | | | 2 |
| 39 | Eucgr.K02223.1 | RBCS2 | 2 | | | |
| 40 | Eucgr.K02765.1 | HAD | | | 2 | |
| 41 | Eucgr.L02773.1 | GCT | | | | |

**Fig 6. Most differentially abundant chloroplast proteins isolated from *Eucalyptus grandis* leaves in studies seasons.** Red boxes indicate proteins with significant change higher than 2-folds. Blue boxes indicate a significant change higher than 5-folds. White boxes indicate proteins with change lower than 2-folds.

In conclusion, our findings suggest that transition seasons (spring and fall) induce the most pronounced chloroplast proteome changes over the year. Out of the 431 proteins isolated in the chloroplast proteome, 41 showed significant abundance changes in different seasons. We also found distinct patterns of protein abundance among the seasons and were able to pinpoint the most abundant proteins in both the dry and wet periods of the year.

## The importance of studying chloroplasts metabolic changes

Chloroplast biogenesis starts with proplastids that are exposed to light and then transformed into mature chloroplasts, which differentiate from gerontoplasts during senescence. Leaf senescence can be triggered by both external and internal factors. Internal factors include age, reproductive growth, redox metabolism, and hormone levels [75–77]. Leaf premature senescence can cause productivity losses due to lower assimilatory capacity [78]. Investigating the complex molecular network behind the developmental processes is essential for understanding the continual transition of functional and regulatory metabolic pathways occurring multiple times throughout the perennials plants lifespan. It is also quite relevant for agriculture because [79, 80] it may open new possibilities to increase crop yield and establish strategies to maximize biomass production.

An inter-organellar transcriptomic analysis of leaf senescence in *Arabidopsis* indicated that the chloroplast transcriptome exhibited the most pronounced changes throughout the plant lifespan [81]. These findings indicate that plastid compartment could be an important target to plant breeding programs that intend to delay leaf senescence and maximize biomass production. Regarding chloroplast metabolism, a large number of differentially expressed transcripts were identified in mature to senescence stages, and the biological processes associated with biogenesis at the growth to maturation stage showed higher expression, suggesting robust transcriptional coordination during early leaf development and more dynamic biochemical transitions during the senescence process [81]. At the protein level, we also observed higher abundance in biogenesis processes in young leaf tissue; in contrast, we detected a larger number of differential abundance proteins for the young stage rather than in mature tissues. This indicates that protein-level dynamics diverges from transcriptome dynamics. The plants also presented a strong translational coordination during early leaf development, and such biochemical networks seem more active during early stages.

Plants are sessile organisms and need to develop survival strategies due to changing environmental conditions. Perennial plants have been described to be strongly affected by seasonally varying availability of nutrients, water, light, and temperatures [30, 31]. Seasonal climate variations can drive plant growth and development, favoring a metabolic adaptation of the life cycle with changes in the environment [29].

Our findings suggest that transition seasons (spring and fall) induce the most pronounced chloroplast proteome changes over the year in *Eucalyptus grandis* cultivated in tropical climate. For the fall, we found higher abundance of proteins related to epigenetic events that are very sensitive to external environment changes. Such epigenetic mechanisms are among the factors that help plants adapt to different environmental climates [59]. The combination of seasonal proteomics and epigenomics studies in natural environments can be important to reveal seasonal dynamics regulation at each level on plant responses plasticity.

Moreover, agrometeorological data revealed extremely dry autumn and winter seasons and mean monthly rainfall of almost 10 mm over the summer. Photosynthetic organisms have established several strategies to cope with dry environments, including short life cycle, enhanced water uptake and reduced water loss, osmotic adjustment, and antioxidant capacity [82]. *Eucalyptus* species have been reported to have developed a strategy involving lower

photosynthetic activity during dry periods associated with resource reallocation through major changes in the gene expression of primary metabolism [83]. Identifying these metabolic changes allows to further investigate and understand the mechanisms behind plants adaptation and assist tree breeding programs.

## Conclusions

Investigation of *Eucalyptus grandis* chloroplast proteome provided a comprehensive resource for understanding different processes underlying age-dependent and seasonal variation in an organellar level. Our chloroplast extraction and enrichment strategies were effective at reducing cross-contaminants abundance and allowed a deep investigation of the changes occurring in the chloroplast proteome throughout leaf development and the seasons of the year. Distinct patterns of protein accumulation were detected in both assays. The *Eucalyptus grandis* chloroplast proteome seems to increase metabolisms related to catabolic and redox processes as leaves age, while the accumulation of proteins involved in biosynthetic processes are downregulated. Most pronounced proteome changes occurred in the transition seasons of the year with a clear proteome bias towards photosynthesis over the wet seasons.

A deeper understanding on age-associated biological networks is important for developing strategies to maximize the biomass production. Furthermore, analyzing seasonal dynamics unravels the molecular mechanisms underlying metabolism plasticity and plant adaptation in spatiotemporal networks.

## Supporting information

**S1 Fig.** *Eucalyptus grandis* **sampling.** Leaf developmental assay, branches were divided into 3 different regions (young, middle, and mature), according to the fluorescence data (FV/FM) and chlorophyll relative quantification (CCI) (A). Proteome seasonal variation assay, forty leaves were collected from the first until the fifth node in all four seasons (Spring, Summer, Fall and Winter) (B).
(TIFF)

**S2 Fig.** *Eucalyptus grandis* **leaf responses.** Chlorophyll relative quantification (CCI) (A) and Quantum efficiency of photosystem II (FV/FM) (B). Leaves were isolated from young, middle, and mature regions as described in the Material and Methods section. Different letters indicate significant differences according to Tukey's test (p <0.05).
(TIF)

**S1 Table. Meteorological data recorded at the Sao Paulo State University Forest Nursery during** *Eucalyptus grandis* **cultivation.**
(XLSX)

**S2 Table.** *Eucalyptus grandis* **non-redundant protein dataset.**
(XLSX)

**S3 Table. Expression profile of the differentially regulated proteins identified in the** *Eucalyptus grandis* **chloroplasts during leaf development.** Proteins were classified according to their abundance profile: "upregulated"—proteins with the highest abundance in mature leaves; "downregulated"–proteins with the lowest abundance in young leaves; "undefined"–proteins with undefined profile pattern.
(DOCX)

**S4 Table. Differentially abundant proteins according to the Tukey's test (p <0.05) in** *Eucalyptus grandis* **chloroplasts isolated in different seasons of the year classified per protein**

**expression pattern.**
(XLSX)

## Author Contributions

**Conceptualization:** Tiago Santana Balbuena.

**Formal analysis:** Amanda Cristina Baldassi.

**Funding acquisition:** Tiago Santana Balbuena.

**Investigation:** Amanda Cristina Baldassi, Tiago Santana Balbuena.

**Methodology:** Amanda Cristina Baldassi.

**Project administration:** Tiago Santana Balbuena.

**Supervision:** Tiago Santana Balbuena.

**Visualization:** Amanda Cristina Baldassi.

**Writing – original draft:** Amanda Cristina Baldassi, Tiago Santana Balbuena.

**Writing – review & editing:** Tiago Santana Balbuena.

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
