## [Decision Letter · Decision Letter 0]

24 May 2022

PONE-D-22-05412The Eucalyptus grandis chloroplast proteome: leaf development and seasonal variationsPLOS ONE

Dear Dr. Balbuena,

Thank you for submitting your manuscript to PLOS ONE. After careful consideration, we feel that it has merit but does not fully meet PLOS ONE’s publication criteria as it currently stands. Therefore, we invite you to submit a revised version of the manuscript that addresses the points raised during the review process.

We look forward to receiving your revised manuscript.

Kind regards,

Mayank Gururani

Academic Editor

PLOS ONE

Journal Requirements:

This work had financial support from São Paulo Research Foundation - FAPESP (grant number 2018/15035-8). ACB also received a scholarship from the same foundation (scholarship number 2019/12580-8). TSB received a scholarship from CNPq (304479/2020-9).

No authors have competing interests

Reviewers' comments:

Reviewer's Responses to Questions

**Comments to the Author**

1. Is the manuscript technically sound, and do the data support the conclusions?

Reviewer #1: Partly

Reviewer #2: Yes

2. Has the statistical analysis been performed appropriately and rigorously? 

Reviewer #1: Yes

Reviewer #2: Yes

3. Have the authors made all data underlying the findings in their manuscript fully available?

Reviewer #1: Yes

Reviewer #2: Yes

4. Is the manuscript presented in an intelligible fashion and written in standard English?

Reviewer #1: No

Reviewer #2: No

5. Review Comments to the Author

Reviewer #1: The chloroplast proteome is encoded by both nuclear genome and its own genome, which play critical roles in plant photosynthesis, metabolism and other biological processes. In recent years, characterization of chloroplast proteome changes under environmental stress condition has attracted the attention of the plant biologists. Correspondingly, the topic of this manuscript is of interest. The work seems to be well done and the results carefully described. Overall, the figures and descriptions are good. However, there are several major problems and many mistakes. I do not find this manuscript publication-ready at this stage for the following reasons:

Major concerns

[1] A major drawback of this study is the purity of the chloroplast fraction. On what basis the authors made the statement “…...that our chloroplast isolation and enrichment strategy was effective in reducing the number of cross-contaminants…...” (Results and Discussion: Line 259-262; Final Remark: Line 679-682). Use of NSAF approach for determining organelle abundance typically rely on detection of a very few marker proteins, and hence is unsatisfactory. In my opinion, there should be a few more more wet-lab experiments that would justify the findings of proteomics analysis. The authors should pick up a few candidates and examine their translocation the chloroplasts, say by transient expression analysis.

[2] The manuscript describes some interesting aspects of biochemistry operating in chloroplast proteome. The authors’ preparation of chloroplast yields a considerable number of unexpected proteins, which are classically considered to be translocated to other cellular compartments. Since contamination from other organelles cannot be discounted, the information obtained for the unexpected proteins in the fraction remains highly suspected. The authors should consider revising the manuscript by focusing on chloroplast proteins, and reducing the description and discussion of the unexpected proteins to a minimum.

[3] The authors stated, in the abstract, that “This study contributes to a more comprehensive understanding on the subcellular mechanisms that lead to plant leaf adaptation and ultimately to Eucalyptus grandis productivity”. There is a need to revisit the statement because it is not convincing in the case of Eucalyptus grandis productivity.

[4] Eucalyptus leaves are a great source of antioxidants, particularly flavonoids, which are effective in protection against oxidative stress and ROS damage. I didn’t find any mention of this sort in the manuscript. Instead, the first two paragraphs of the Introduction contain too much general information which are text book materials. I feel that these paragraphs can be clubbed and restructured. Noticeably, I have counted as many as 41 references in the Introduction, many of which describe the general aspects of chloroplast biogenesis and the function of chloroplast in plant. The number of references in this section should be reduced.

[5] “…… 50 μg of proteins were separated for 1 hour by denaturing electrophoresis on a polyacrylamide gel containing sodium dodecyl sulfate (SDS-PAGE) to obtain a single band”. What is the meaning of “to obtain a single band”?

[6] In general, experimental protein localization requires a great deal of time and resources, which are bypassed by prediction analysis. It is recommended that the prediction analysis be accomplished by a series of prediction tools for cross-validation test. In this study, subcellular localization of the proteins was predicted by ChloroP and Predotar. Why don’t add some other tools, for example, BaCelLo, WoLF PSORT, TargetP etc.

Minor concerns

Authors should ensure the text is polished enough to be published. The English needs to be much improved and the authors should request the assistance of a professional service. While the entire text requires revision, I will cite some issues that should be addressed to avoid errors and misinterpretations. Please note that the revision should not be limited to the following items:

[1] Please carefully edited the manuscript; remove most acronyms unless very much necessary.

[2] “…...may contribute to enlarge the understanding…...” (Abstract) is not a good read. Similarly, “…… proteome analysis of the Eucalyptus grandis chloroplast proteome….”. These should be suitably rephrased.

[3] There must be a space between the numerical value and unit symbol; missing at so many places in the text.

[4] “Eucalyptus grandis were planted” is more appropriate than “Eucalyptus grandis plants were cultivated” (Materials and Methods: Line 108).

[5] The sentence “For both assays using leaf sampling, plant material………chloroplasts, as described in the next sections” (Line 124-125) is absolutely unnecessary.

[6] Rephrase the sentence “Fluorescence analysis was performed on the fluorometer FluorPen FP …………” (Line 130-132).

[7] “Aliquots of six grams of leaves...” is imperfect in the sentence (Line 143-144).

[8] “……seems prone to catabolic and redox processes as leaves age….” (Line 683-684) doesn’t make sense.

[9] Fig. S2 legend: Which one is correct: chlorophyll relative quantification (CCI) or chlorophyll content index (CCI)?

[10] “up-regulated” and “down-regulated” can be replaced with “upregulated” and “downregulated”, respectively throughout the text.

[11] References need extensive corrections since the references are not in the same format [as per the style of PLoS ONE], particularly the journal names (Reference 33: Environ. Sci. Pollut. Res.; Reference 33: J Exp Bot.).

Reviewer #2: The manuscript describes proteomics changes in chloroplast organelle of Eucalyptus grandis during development and in seasonal changes. I found it very interesting since it adds a brick into the wall of knowledge of plant metabolism changes in physiological conditions providing useful information on functional networks at the basis of development process. Further, it is well known that chloroplast plays a central role in plant metabolism and thus in adaptation to environmental changes, thus investigation of plastid proteome may help in understanding plant adaptation mechanisms that can be exploited to increase plant productivity.

Overall, the manuscript is clear, however, I have some suggestions.

I have some concern on the possibility to excise a single band from an SDS-PAGE gel where 50 ug of proteins have been loaded. Often, using a 2D-PAGE overlapping of bands occurs.

I invite the authors to be consistent with Figs and Tables’ nomenclature (e.g., 2S vs S2)

Figure S2B is a duplication of S1A

In all the text many words are collapsed with spaces missing (e.g., L59, 64, 70…304, 351, 518, 527, 550 etc)

L341-63 and 440-53, please, insert the full protein name (as authors do in L413-419)

L352 and 451 “PRPL1” is a typo?

L424-425 this sentence is not clear

L413-19 and 433-37 short name of proteins should be near their full name

Please, pay attention to typos (e.g., L580 “the” is duplicated, “fakk” should be “fall”)

6. PLOS authors have the option to publish the peer review history of their article (what does this mean?). If published, this will include your full peer review and any attached files.

Reviewer #1: **Yes: **Dr. Niranjan Chakraborty

Reviewer #2: No

---

## [Author Response · Author response to Decision Letter 0]

13 Jun 2022

The Eucalyptus grandis chloroplast proteome: leaf development and seasonal variations

Amanda Cristina Baldassi and Tiago Santana Balbuena

Response to Reviewers

Reviewer #1

Major concerns

[1] A major drawback of this study is the purity of the chloroplast fraction. On what basis the authors made the statement “…...that our chloroplast isolation and enrichment strategy was effective in reducing the number of cross-contaminants…...” (Results and Discussion: Line 259-262; Final Remark: Line 679-682). Use of NSAF approach for determining organelle abundance typically rely on detection of a very few marker proteins, and hence is unsatisfactory. In my opinion, there should be a few more more wet-lab experiments that would justify the findings of proteomics analysis. The authors should pick up a few candidates and examine their translocation the chloroplasts, say by transient expression analysis.

Response:We understand the concerns regarding this issue. In addition to the data described in the manuscript, the supplementary material 2 of one of our group’s publication (https://doi.org/10.1007/s00468-018-1750-8) gives support to our chloroplast isolation procedure. We did not carry out a Western Blot Immuno assays to confirm results as the fragment ions from MS/MS data and stringency used for protein identification give support for the identifications proposed in the manuscript. We also used 2 different quantification metrics (NSAF and LFQ intensity) to confirm the decrease in abundance of the contaminants. However, we agree that the sentence “…...that our chloroplast isolation and enrichment strategy was effective in reducing the number of cross-contaminants…...” have to berephrased since Table 1 showed reduction in the abundance of the contaminants but not in the number contaminant IDs.

[2] The manuscript describes some interesting aspects of biochemistry operating in chloroplast proteome. The authors’ preparation of chloroplast yields a considerable number of unexpected proteins, which are classically considered to be translocated to other cellular compartments. Since contamination from other organelles cannot be discounted, the information obtained for the unexpected proteins in the fraction remains highly suspected. The authors should consider revising the manuscript by focusing on chloroplast proteins, and reducing the description and discussion of the unexpected proteins to a minimum.

Response:Reviewer is correct, even though we used a chloroplast isolation procedure, we indeed identified a considerable number of unexpected proteins. However, it should be highlighted that we only discussed data from proteins whose cellular location was predicted to be chloroplatic. Identified proteins that were not predicted to be localized in the chloroplast were not taken into consideration in our analysis and discussion. .

[3] The authors stated, in the abstract, that “This study contributes to a more comprehensive understanding on the subcellular mechanisms that lead to plant leaf adaptation and ultimately to Eucalyptus grandis productivity”. There is a need to revisit the statement because it is not convincing in the case of Eucalyptus grandis productivity.

Response: Reviewer is correct. The statement was rephrased. 

[4] Eucalyptus leaves are a great source of antioxidants, particularly flavonoids, which are effective in protection against oxidative stress and ROS damage. I didn’t find any mention of this sort in the manuscript. Instead, the first two paragraphs of the Introduction contain too much general information which are text book materials. I feel that these paragraphs can be clubbed and restructured. Noticeably, I have counted as many as 41 references in the Introduction, many of which describe the general aspects of chloroplast biogenesis and the function of chloroplast in plant. The number of references in this section should be reduced.

Response:Reviewer is correct. The paragraphs were restructured. 

[5] “…… 50 μg of proteins were separated for 1 hour by denaturing electrophoresis on a polyacrylamide gel containing sodium dodecyl sulfate (SDS-PAGE) to obtain a single band”. What is the meaning of “to obtain a single band”?

Response: The goal of loading the chloroplast extracts on SDS-Page gel was to clean up samples from salts, denaturants, detergents and other buffer components that could interfere in the MS analysis. We only allowed 1 cm of samples to enter into the gel and excised the entire segment as a single band as you can see in the picture below:

[6] In general, experimental protein localization requires a great deal of time and resources, which are bypassed by prediction analysis. It is recommended that the prediction analysis be accomplished by a series of prediction tools for cross-validation test. In this study, subcellular localization of the proteins was predicted by ChloroP and Predotar. Why don’t add some other tools, for example, BaCelLo, WoLF PSORT, TargetP etc.

Response: We usedtwodifferent prediction tools: ChloroP that uses chloroplast transit peptide and Predotar that uses N-terminal targeting sequences to predict chloroplast proteins. Instead of adding other prediction tools into the pipeline, we combined data from those tools with data from chloroplast protein sequence databases (PPDB and AT-Chloro) in order to identify the chloroplast proteins described in the manuscript. Although not used in our research, we evaluated the potential gains of adding other prediction tools. As we can see below, contribution of other prediction tools is limited to approximately 10% of the current identifications and, thus, were not incorporated into our work.

Minor concerns

Authors should ensure the text is polished enough to be published. The English needs to be much improved and the authors should request the assistance of a professional service. While the entire text requires revision, I will cite some issues that should be addressed to avoid errors and misinterpretations. Please note that the revision should not be limited to the following items:

[1] Please carefully edited the manuscript; remove most acronyms unless very much necessary.

Response: We removed them.

[2] “…...may contribute to enlarge the understanding…...” (Abstract) is not a good read. Similarly, “…… proteome analysis of the Eucalyptus grandis chloroplast proteome….”. These should be suitably rephrased.

Response: We did it.

[3] There must be a space between the numerical value and unit symbol; missing at so many places in the text.

Response: We corrected it.

[4] “Eucalyptus grandis were planted” is more appropriate than “Eucalyptus grandis plants were cultivated” (Materials and Methods: Line 108).

Response: We corrected it.

[5] The sentence “For both assays using leaf sampling, plant material………chloroplasts, as described in the next sections” (Line 124-125) is absolutely unnecessary.

Response: We removed the sentence.

[6] Rephrase the sentence “Fluorescence analysis was performed on the fluorometer FluorPen FP …………” (Line 130-132).

Response: We rephrased accordingly.

[7] “Aliquots of six grams of leaves...” is imperfect in the sentence (Line 143-144).

Response: We corrected it.

[8] “……seems prone to catabolic and redox processes as leaves age….” (Line 683-684) doesn’t make sense.

Response: We corrected it.

[9] Fig. S2 legend: Which one is correct: chlorophyll relative quantification (CCI) or chlorophyll content index (CCI)?

Response:Chlorophyll relative quantification is correct.We corrected in the manuscript.

[10] “up-regulated” and “down-regulated” can be replaced with “upregulated” and “downregulated”, respectively throughout the text.

Response: We replaced the term throughout the text.

[11] References need extensive corrections since the references are not in the same format [as per the style of PLoS ONE], particularly the journal names (Reference 33: Environ. Sci. Pollut. Res.; Reference 33: J Exp Bot.).

Response: References were revised and corrected.

Reviewer #2

I have some concern on the possibility to excise a single band from an SDS-PAGE gel where 50 ug of proteins have been loaded. Often, using a 2D-PAGE overlapping of bands occurs.

Response:As we answered to the reviewer #1, the goal of loading the chloroplast extracts on SDS-PAGE gel was to clean up samples from salts, denaturants, detergents and other buffer components that could interfere in the MS analysis.

I invite the authors to be consistent with Figs and Tables’ nomenclature (e.g., 2S vs S2)

Response: We did it. 

Figure S2B is a duplication of S1A

Response: The figure S2 was modified. We decided to keep only S2A with CCI and FV/FM results. 

In all the text many words are collapsed with spaces missing (e.g., L59, 64, 70…304, 351, 518, 527, 550 etc)

Response:True. We revised this issue throughout the text.

L341-63 and 440-53, please, insert the full protein name (as authors do in L413-419)

Response:The full protein names were addedin the Supplementary material (S3 Table).

L352 and 451 “PRPL1” is a typo?

Response:True. Thanks for pointing this out. Werevised this issue throughout the text.

L424-425 this sentence is not clear

Response:We tried to clarify the sentence.

L413-19 and 433-37 short name of proteins should be near their full name

Response:We corrected this issue.

Please, pay attention to typos (e.g., L580 “the” is duplicated, “fakk” should be “fall”)

Response:We revised the manuscript in order to find and correct typos.

---

## [Decision Letter · Decision Letter 1]

12 Jul 2022

PONE-D-22-05412R1The Eucalyptus grandis chloroplast proteome: leaf development and seasonal variationsPLOS ONE

Dear Dr. Balbuena,

Thank you for submitting your manuscript to PLOS ONE. One of the reviewers has accepted your MS while the second reviewer has suggested some minor changes. 

We look forward to receiving your revised manuscript.

Kind regards,

Mayank Gururani

Academic Editor

PLOS ONE

Journal Requirements:

Reviewers' comments:

Reviewer's Responses to Questions

**Comments to the Author**

1. If the authors have adequately addressed your comments raised in a previous round of review and you feel that this manuscript is now acceptable for publication, you may indicate that here to bypass the “Comments to the Author” section, enter your conflict of interest statement in the “Confidential to Editor” section, and submit your "Accept" recommendation.

Reviewer #1: (No Response)

Reviewer #3: All comments have been addressed

2. Is the manuscript technically sound, and do the data support the conclusions?

Reviewer #1: (No Response)

Reviewer #3: Yes

3. Has the statistical analysis been performed appropriately and rigorously? 

Reviewer #1: (No Response)

Reviewer #3: Yes

4. Have the authors made all data underlying the findings in their manuscript fully available?

Reviewer #1: (No Response)

Reviewer #3: Yes

5. Is the manuscript presented in an intelligible fashion and written in standard English?

Reviewer #1: (No Response)

Reviewer #3: Yes

6. Review Comments to the Author

Reviewer #1: Compared to the original manuscript, the revised version has been greatly improved, especially with respect to structural rearrangement and copyediting. The authors addressed most of the reviewers' questions and comments. Yet I have few concerns which are listed below:

[1]. I would prefer a modified title “The Eucalyptus grandis chloroplast proteome: seasonal variations in leaf development” and over “The Eucalyptus grandis chloroplast proteome: leaf development and seasonal variations over”.

[2]. The last line in the abstract “Mass spectrometric data are available via ProteomeXchange under identifier PXD029004” can be moved to “Results and Discussion” section.

[3]. “Transient seasons” in the subtitle “Transient seasons induce the most pronounced chloroplast

proteome changes” is not a good read and should be modified.

[4]. The phrase “Proteome isolation” is not appropriate and hence the subtitles “Chloroplast proteome isolation” and “Total leaf proteome isolation” should be modified.

[5]. Subtitle “Final Remarks” can be replaced with “Conclusions”.

Reviewer #3: After going through the revised MS and the replies to the reviewer's queries, I believe that the MS has already been critically evaluated by the previous reviewers and that pertinent questions were asked by the reviewers. In response to their queries, the authors have addressed all the concerns adequately which has significantly improved the scientific merit of the article. Based on these observations, I believe that the MS can be accepted in its current form.

7. PLOS authors have the option to publish the peer review history of their article (what does this mean?). If published, this will include your full peer review and any attached files.

Reviewer #1: **Yes: **Niranjan Chakraborty

Reviewer #3: No

---

## [Author Response · Author response to Decision Letter 1]

15 Aug 2022

RESPONSE TO REVIEWERS

Reviewer #1: 

Compared to the original manuscript, the revised version has been greatly improved, especially with respect to structural rearrangement and copyediting. The authors addressed most of the reviewers' questions and comments. Yet I have few concerns which are listed below:

[1]. I would prefer a modified title “The Eucalyptus grandis chloroplast proteome: seasonal variations in leaf development” and over “The Eucalyptus grandis chloroplast proteome: leaf development and seasonal variations over”.

>>Authors: We changed the MS title according to this suggestion.

[2]. The last line in the abstract “Mass spectrometric data are available via ProteomeXchange under identifier PXD029004” can be moved to “Results and Discussion” section.

>>Authors: We moved the sentence to the “Results and Discussion” section.

[3]. “Transient seasons” in the subtitle “Transient seasons induce the most pronounced chloroplast proteome changes” is not a good read and should be modified.

>>Authors: We incorporated this change.

 [4]. The phrase “Proteome isolation” is not appropriate and hence the subtitles “Chloroplast proteome isolation” and “Total leaf proteome isolation” should be modified.

>>Authors: True. Thanks for pointing this out. We changed the phrase.

[5]. Subtitle “Final Remarks” can be replaced with “Conclusions”.

>>Authors: We incorporated this change.

We thank both Reviewers for their suggestions and critics. Thank you for using your time and energy to make it a better manuscript.

---

## [Editor Report · Decision Letter 2]

18 Aug 2022

The Eucalyptus grandis chloroplast proteome: seasonal variations in leaf development

PONE-D-22-05412R2

Dear Dr. Balbuena,

We’re pleased to inform you that your manuscript has been judged scientifically suitable for publication and will be formally accepted for publication once it meets all outstanding technical requirements.

Kind regards,

Mayank Gururani

Academic Editor

PLOS ONE
---

## [Editor Report · Acceptance letter]

25 Aug 2022

PONE-D-22-05412R2 

The *Eucalyptus grandis* chloroplast proteome: seasonal variations in leaf development 

Dear Dr. Balbuena:

I'm pleased to inform you that your manuscript has been deemed suitable for publication in PLOS ONE. Congratulations! Your manuscript is now with our production department. 

Kind regards, 

on behalf of

Dr. Mayank Gururani 

Academic Editor

PLOS ONE